# Effectiveness of Warm-Up Routine on the Ankle Injuries Prevention in Young Female Basketball Players: A Randomized Controlled Trial

**DOI:** 10.3390/medicina55100690

**Published:** 2019-10-16

**Authors:** Elvira Padua, Agata Grazia D’Amico, Anas Alashram, Francesca Campoli, Cristian Romagnoli, Mauro Lombardo, Matteo Quarantelli, Emanuele Di Pinti, Christian Tonanzi, Giuseppe Annino

**Affiliations:** 1Department of Human Science and Promotion of Quality of Life, San Raffaele Open University of Rome, 00166 Rome, Italyfrancesca.campoli@uniroma5.it (F.C.); mauro.lombardo@uniroma5.it (M.L.); matteo.quarantelli@uniroma5.it (M.Q.); 2School of Neuroscience, Faculty of Medicine and Surgery, University of Rome Tor Vergata, 00133 Rome, Italy; anasalashram@gmail.com; 3Departement for Life Quality Studies, Alma Mater Studiorum University of Bologna, 47921 Rimini, Italy; cristian.romagnoli2@unibo.it; 4Piuvistamedsport, 19085 Villanova (Rome), Italy; emanuele.dipinti@gmail.com; 5OsteoMov, 00146 Rome, Italy; c.tonanzi@hotmail.it; 6Department of Medicine Systems, University of Rome, “Tor Vergata”, 00133 Rome, Italy; g_annino@hotmail.com

**Keywords:** ankle injury, basketball players, core stability, dorsiflexion, injury prevention

## Abstract

*Background and Objectives*: Ankle joint is the most common site of injury for basketball athletes. An effective warm-up (WU) is a period of preparatory exercise to improve training performance and reduce sports injuries. Continuous examination of effective WU routines in basketball players is a necessity. The aim of this study was to investigate the effects of general and combined warm up on ankle injury range of motion (ROM) and balance in young female basketball players. *Materials and Methods*: A sample of 28 young female basketball players were randomly allocated to either global warm up control group (GWU) (*n* = 11) or combined warm up experimental group (CWU) (*n* = 17). All participants performed 7-min of run. The CWU group performed a single leg stance barefoot with eyes closed, plank forearm position and triceps sural stretching. Participants in GWU performed walking ball handling and core stability using a Swiss ball. Both WU routines were conducted 3 times per week for 10 weeks. Outcome measurements were the Stabilometric platform and dorsiflexion lunge test. *Results*: Twenty-eight young female basketball players completed the study. Participants in the experimental group improved significantly in the range of motion (ROM) in right and left ankle and the center of pressure displacement (CoP). The control group did not show any changes in ankle dorsiflexion and a significant reduction in all body balance parameters. *Conclusions*: An 8-min combined warm-up routine for 10 weeks improves the ankle dorsiflexion ROM and CoP displacement that plays a key role in ankle injuries prevention in basketball players. Further studies are strongly needed to verify our findings.

## 1. Introduction

The ankle joint is the most common site of injury in athletes [1]. Ankle sprains account for 76.7% of injuries, followed by ankle fractures 16.3%. Up to 80% of athletes with ankle sprain will suffer recurrent sprains, and up to 72% develop chronic instability [2]. Basketball has a higher rate of ankle injuries especially between the female athletes [1]. Basketball athletes are five times more likely to injure an ankle joint after a prior ankle injury, with a recurrence rate of 73% [3,4]. Ankle injuries are considered the most common and severe impairments affecting lower extremities [5,6]. Athletes with ankle injuries experiencing disability and residual symptoms such as pain, sense of instability, crepitus and weakness [7,8]. It is therefore important that ankle injuries are reduced to ensure basketball players can continue to maintain their sport participation and performance.

Athletes who did not perform a general stretching as part of their warm-up protocol were 2.6 times more likely to have ankle injury than athletes who stretched [9]. After an ankle sprain, low ankle dorsiflexion ROM of ankle joint is a risk factor for developing patellar tendinopathy in basketball players [10] as well as impaired bilateral balance [11]. This loss of function increases the risk of re-sprains [10]. Proper warm-up before starting the exercises or practice routine, including ankle joint mobility and balance training, substantially reduces the risk of sustaining ankle sprains; this effect is seen especially in those with a history of previous ankle sprains [12,13].

Warm-up (WU) is defined as a period of preparatory exercise performed by athletes in order to improve neuromuscular performance and reduce sports injuries [14]. In fact, WU increases performance through reducing the viscous resistance of muscles (i.e., smoother contraction), and increases the speed of nerve transmission [15,16,17,18]. In addition, balance training also contributes to improvement in motor skills, resulting in an increase of force development rate and therefore, improved athletic performance [19].

Some studies have manipulated warm up duration [20,21,22,23] and/or intensity [24,25,26] in order to increase muscle temperature and performance with contradictory results. Despite some studies reported an increase in explosive performance after general [27], specific [28,29] and combined WU [13], others have not reported a significant effect [27,30,31,32]. Moreover, specific warm-up effects focused on ankle joint mobility and balance exercise routine have been poorly studied [33,34,35]. Thus, the aim of this study was to investigate if a warm up focused on ankle joint mobility and body balance could be effective to increase these abilities in order to prevent ankle sprain injuries in young female basketball players.

## 2. Materials and Methods

### 2.1. Participants

Twenty-eight young female basketball players in (MSC Basketball Academy, Rome, Italy) participated in the study after an informed consent approved from their parents. Participants were homogenous regarding a competitive level ranging from 2 to 3 years of practice, which they trained at least 3 times per week. The exclusion criteria were; recent history of muscle injuries, trauma or any other illness that could compromise their daily training. The study was approved by the Institutional Research Board of the University online San Raffaele of Rome, Italy. All procedures were carried out in accordance with the Declaration of Helsinki. Table 1 presents participant’s characteristics.

### 2.2. Study Design and Intervention

The Participants who met the inclusion criteria were 28 participants. They were randomly assigned to either a combined warm-up experimental group (CWU) or general warm-up control group (GWU) using a computer-generated randomization list. In order to prevent any unexpected logistic problems related to the young age, three players previously assigned to a control group was added to the experimental group changing the sample size of both groups (CMU = 17 players, GWU = 11 players).

Participants in both experimental and control groups underwent a 10-week warm-up, 3 times/weekand performing run with different gaits (7 min) (Figure 1). After that, participants in the experimental (CWU) group underwent combined warm-up routine; balance training (2 min) including; performing a single leg stance barefoot with eyes closed (four tasks of 25 s for each limb); core stability performing a plank forearm position (4 series of 25 s) and dorsiflexion ankle mobility using the dorsiflexion long position for the triceps sural stretching (1-min for each limb). Participant in control (GWU) group underwent a standard warm-up routine; walking ball handing (2 min), core Stability using Swiss ball (6 min). After the warm-up, participants in both groups spent the same amount of time on technical and tactical basketball practice.

### 2.3. Outcome Measures

All participants were assessed at baseline and reassessed at the end of treatment by one assessor blinded to the intervention. The main outcome was the Stabilometric platform (Free Step Sensor Medica). The secondary outcome was the dorsiflexion long test (DLT).

Stabilometric platform (Free Step Sensor Medica) [34] assessed the balance status of the postural tone. It registers the vertical forces exerted by the weight force of the human body and evaluated through the measurement of the amplitude of center of pressure (CoP), performed with open eyes, calculating the number of body sways for each subject. The medium-lateral and anterior-posterior fluctuations and the total distance of the oscillation (total length of the CoP line) were registered. This measurement represents the energy spending state of the subject in maintaining the orthostatic position. No warning regarding posture was provided to the participants.

Dorsiflexion long test (DLT) represents a quantitative measurement used to estimate exclusively the ankle mobility [36]. It was performed barefoot and it was asked to the participants to place the heel center of tested foot perpendicular to the wall the first toe located in a marked line on the ground by a piece of guide tape. Then, the participants were asked to flex their knees until just to touch the wall, reaching the maximum ankle dorsiflexion, and measuring the maximum distance from the wall to the toe without lifting the examined heel. Three measures were taken for each leg and the mean used for data analysis [37]. In basketball, the adequate values of ankle dorsiflexion are set about 8 cm equivalent to 36.5° and the difference between the two limbs, left and right, must be around 1.5 cm [38].

### 2.4. Statistical Analysis

Mann-Whitney U test was used to compare differences among groups; the difference in outcome measures within each group was assessed by the Wilcoxon Signed Ranks test. The level of significance for all statistical tests was set at *p* ≤ 0.05. Effect sizes were measured to determine the differences between pre-test and post-test values within the group using the formula of r = Z/√N (Z: Z-value, N: number of observations). Effect size considered small at 0.1, moderate at 0.3 and large at 0.5 [37]. Statistical analysis was conducted using SPSS statistics version 24.

## 3. Results

The anthropometric characteristics of the participants are reported in Table 1. No significant differences were found between both groups in demographic characteristics and in outcome measures at baseline.

Right ankle dorsiflexion improved in experimental group (*p* < 0.001, r = 0.61), no improvement was reported in control group (*p* = 0.11, r = 0.34) (Figure 2A). There was significant improvement in left ankle dorsiflexion (*p* < 0.05, r = 0.57) in experimental group, no improvement was reported in control group (*p* = 0.12, r = 0.34) (Figure 2B).

Regarding the balance test, experimental group showed significant reduction in center of pressure (CoP) displacement (*p* < 0.05 r = 0.50), while control group showed significant increase (*p* < 0.05, r = 0.60) (Figure 3A). The surface epill (single-limb stance on the stabilometric platform) increased significantly in the control group (*p* < 0.05 r = 0.53) only, while the experimental group did not show any significant changes (*p* = 0.83, r = 0.04) (Figure 3B). With the exception of pathway length, there were no significant differences between groups in any of the outcome measures at the end of warm-up.

## 4. Discussion

To our knowledge, this is the first study evaluated the efficacy of ankle mobility and balance exercises in the warm-up routine on ankle injury prevention in basketball female players. Results showed significant improvement in both ankle dorsiflexion range of motion and body balance in combined warm-up group compared to the general warm-up group.

Significant reduction in Cop displacement was reported in the experimental group after a combined warm-up routine, while the control group performed a general warm-up routine showed a reduction in balance ability after the global warm-up routine. We propose that the reduction in balance ability in the control group may due to lack of dorsiflexion ankle mobility exercises. The improvements in ankle dorsiflexion after a combined warm-up routine may result from stretching of triceps sural [39,40,41]. In this context some studies showed that the improvement of ankle dorsiflexion ROM is related with a reduced risk of ankle sprains and lower limbs injuries [42,43,44]. Ankle impairment is the main injury affecting basketball players that leads to altered dorsiflexion, which is probably associated with a more erect landing posture and greater ground reaction force [38,45,46]. The compromised dorsiflexion is strictly related to reducing of sports injury, noteworthy the alteration of ankle dorsiflexion was demonstrated following ankle damage [10,47]. The ankle dorsiflexion represents a needful movement to allow normal activity such as walking [48]. In particular, dorsiflexion weight-bearing activities is pivotal in a physical activity besides in sports [49]. This action consists of the contemporary movement of the talus that rolls forward the leg and slides posteriorly. Moreover, in weight-bearing dorsiflexion, a slight tibial happens together talus movement [50]. The assessment of angle dorsiflexion is carried out in various medicine and motor sciences investigations. However, researchers performed different measurements to determine this parameter [46,51]. Then the comparison between different reports appears non-standardized [52]. Among the several alternative methods for measuring ankle dorsiflexion, we have chosen the weight-bearing dorsiflexion since it better reflects the change of position characterizing physical activity [53]. In the present study, we have demonstrated that the training program characterized by proprioceptive exercise is able to significantly improve the dorsiflexion range of motion as compared to the control group.

In this study, the experimental group after a combined warm-up routine group showed a significant decrease in pathway length while the global warm-up routine control group decreased significantly the balance control confirmed also by a significant increase in the ellipse area. According to others studies [52,53] that upright posture tested in barefoot condition on a firm surface is able to reduce the body sway path compared with foam surface as represented by standard sports shoes. In fact, the balance barefoot exercise associated with core stability routine, which performed in experimental intervention significant decreases in body sway path (CoP displacement) compared with the control group. The importance of proprioception has been well established in sports injury prevention and rehabilitation [54]. Probably the interaction between hard gym ground surface and barefoot players conditions, who performed in this study, was enough to maintain the postural control through the activity of the afferent mediated by plantar mechanoreceptors respect to foam surface as used in standard sport shoes [53]. In a recent study, Palazzo and colleagues [55] showed a less decrease of postural control in the group that used textured insoles inside the athletic shoes, compared with the group that used standard athletic shoes. Therefore, it is possible to suggest that the use of balance barefoot exercise routine with a long time could imply a greater amount of information from the foot tactile system with a consequent increase of the postural control and a promising decrease of the sports injury risk.

This study includes some limitations; first, it is not a double-blind study, however, blinded independent assessor to interventions limited risk of bias. Second, the sample size was small, further studies with large sample size are strongly needed to confirm our findings. Third, this study did not contains long follow up, in order to understand the effect of warm-up routines in basketball players. Finally, Future studies should also investigate that the adoption and maintenance of such exercises in warm up necessary to be considered in effectivity trials, including behavioral theory and player and coach perceptions of exercise training [56].

## 5. Conclusions

In conclusion, to insert dorsiflexion long position, core stability and barefoot balance exercises in the training warm-up routine seems to be effective in improving ankle dorsiflexion and Cop displacement in young female basketball players. Further studies should verify our findings including a long follow up.

## Figures and Tables

**Figure 1 medicina-55-00690-f001:**
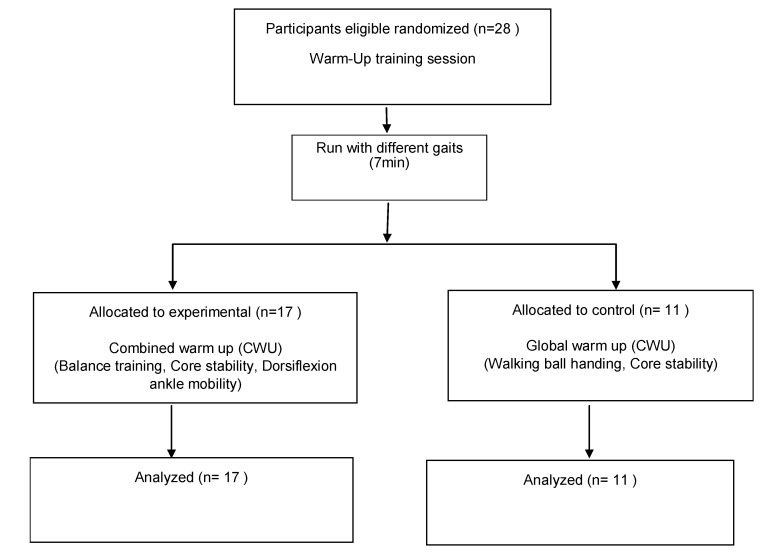
Schematic flow chart of the participants recruited for the study.

**Figure 2 medicina-55-00690-f002:**
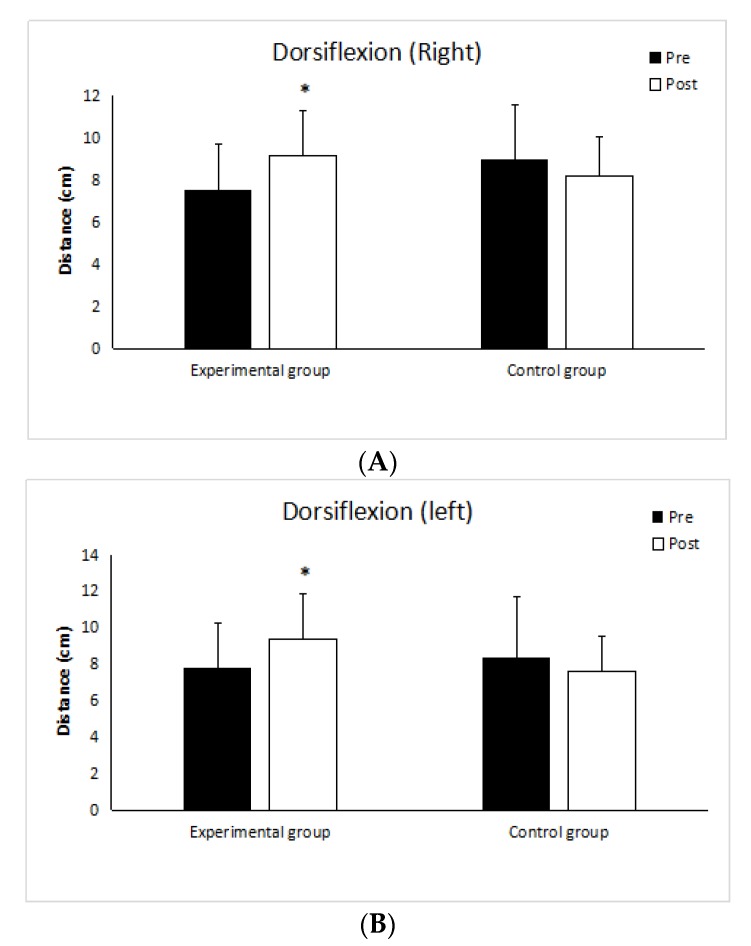
Dorsiflexion long test of right ankle (**A**), dorsiflexion long test of left ankle (**B**). * Significant difference within group *p* < 0.05.

**Figure 3 medicina-55-00690-f003:**
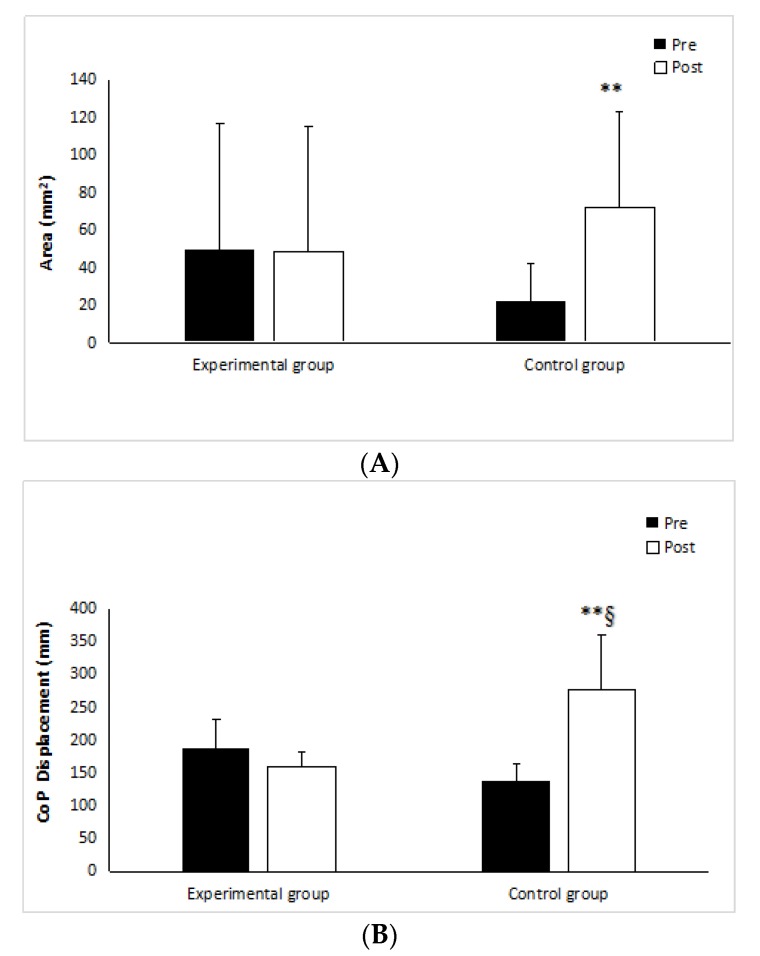
Postural sway measurement on the stabilometric platform (**A**) and the CoP surface (**B**). ** Significant difference within group *p* < 0.01; § significant difference between group *p* < 0.001.

**Table 1 medicina-55-00690-t001:** Characteristics of female basketball athletes recruited to conduct the present study (SD: Standard Deviation).

	Experimental Group (Mean ± SD)	Control Group (Mean ± SD)	All Participants (Mean ± SD)
Age	14.59 ± 1.12	15.44 ± 1.94	14.88 ± 1.48
Weight	55.16 ± 10.78	60.93 ± 12.75	57.43 ± 11.72
Hight	164.32 ± 8.52	163.36 ± 6.86	163.95 ± 7.79
body mass index (BMI)	20.26 ± 2.96	22.70 ± 3.76	21.22 ± 3.45

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
