# Peer review of "Effectiveness of Warm-Up Routine on the Ankle Injuries Prevention in Young Female Basketball Players: A Randomized Controlled Trial"

_medicina, 2019, doi:10.3390/medicina55100690_

Round 1

Reviewer 1 Report

Materials and Methods.

Participants: Please refer to the necessary sample size in order to offer validity of the findings. Table 1: The gender distribution (male, female) is not necessary as the study include only females. 

Outcome measures: Please specify production details of the stabilometric platform. The data regarding the normal values of ankle dorsiflexion quantified by the wall test are not sufficiently sustained. Add more details, references (lines 112-114). 

Results. Provide a table with the results of ankle dorsiflexion and CoP displacement assessments. The term treatment is not adequately used as the intervention refers to the warm up session in the basketball training (line 140).

Discussion. It is clearly explained why the balanced ability decreased in the control group after 10 weeks.

Author Response

Materials and Methods.

Participants: Please refer to the necessary sample size in order to offer validity of the findings. Table 1: The gender distribution (male, female) is not necessary as the study include only females. 

We deleted the row of gender in Table one as the study include only females

Outcome measures: Please specify production details of the stabilometric platform. The data regarding the normal values of ankle dorsiflexion quantified by the wall test are not sufficiently sustained. Add more details, references (lines 112-114). 

We modified it by delete repetition of the sentence. Sufficient information were found in the next sentences within the text

Results. Provide a table with the results of ankle dorsiflexion and CoP displacement assessments. The term treatment is not adequately used as the intervention refers to the warm up session in the basketball training (line 140).

Changed to warm-up instead of treatment

Discussion. It is clearly explained why the balanced ability decreased in the control group after 10 weeks.

We explain why the balance ability decreased in the control group

We propose that the reduction in balance ability in the control group may due to lack of dorsiflexion ankle mobility exercises

Reviewer 2 Report

Comments to the Author

The aim of this study was to investigate if a warm up focused on ankle joint mobility and balance could be effective in ankle sprain injury prevention in young female basketball players. By doing so the Author(s) addressed an important aspect of injury prevention research focusing on specific risk factors for ankle injuries and the efficacy of intervention/s to reduce players risk of ankle injury. Furthermore, the Author(s) wisely decided to explore this area of injury among female basketball athletes for which there is a lack of focus and a major gap in the research. For these reasons the Author(s) should be congratulated for their work.

The paper is generally well written, with a few minor grammatical and expression issues that could be taken into considered for revision. I hope that my comments are helpful to the author(s).

Title

It is recommended that the title of the paper be reworded to be more succinct.

Abstract

Page 1, Line 19-20 – suggest reword sentence for clarity.

Background/objective: Ankle joint is the most common site of injury in overall athletes including basketball. Thus, continuous examination of effective warm-up (WU) routines in 20 basketball players is a necessity. WU is a period of preparatory exercise to improve training 21 performance and reduce sports injuries.

Suggested change - The ankle joint is the most common site of injury for basketball athletes. An effective warm-up (WU) is a period of preparatory exercise to improve training performance and reduce sports injuries. Continuous examination of effective WU routines in basketball players is a necessity.

Page 1, Line 23 – please ensure ROM is spelled out in full with abbreviation for 1st time use.

Page 1, Line 26 – please reword for clarity “All participants performed 7-min of run and sprint”. Is it All participants performed a 7-min run or sprint? Or is it one or the other? Also please check your reference to 7 mins and then 8 mins on the abstract in correct and also consistent with the methodology section.

Page 1, Line 29 - Remove the word ‘used’ at the end of the sentence (WU routines were conducted 3 times per week for 10 weeks used.) , or was there something missing at the end of the sentence? Please ensure this reads clearly.

Page 1, Line 36 - Conclusions – please add wording ‘are’ to ensure sentence is clear. For example,  Further studies are strongly needed to verify the current findings.

Introduction

Page 2, Line 41 – suggest add the word ‘for’ to improve expression ‘Ankle sprains account ‘for’ 76.7% of injuries, followed by ankle fractures 16.3%.

Page 2, Line 47-49 – suggest expression and tense could be improved to enhance clarity and flow of information. Consider the flow in information in introducing the warm up as prevention intervention. Is this sentence placed better in the 2nd paragraph?

Suggest could say something like the following at the end of the 1st paragraph to improve the significance of the research and highlight why it is important, ‘It is therefore important that ankle injuries are reduced to ensure basketball players can continue to maintain their sport participation and performance’.

Suggest in the 1st paragraph could also consider the importance of female injury statistics and that they generally have a higher risk of lower limb injuries than males in basketball. Ensure you highlight why did you focus on female athletes?

Page 2, Line 57 – change ‘facts’ to ‘fact’.

Page 2, Line 59 – change ‘increase’ to ‘increases’

Page 2, Line 63-65 – please improve expression of the following sentence to ensure clarity - ‘Despite some studies reported an increase in explosive performance was reported after general [27] or specific [28,29] or combined WU [13], others have not reported a significant effect [27,30,31,32].

Page 2, Lines 65-67 – please consider if this sentence is entirely true and modify accordingly (including references), many lower limb injury prevention programs have been developed worldwide which include ankle joint mobility and balance exercises in other sports and some applied to basketball such as the FIFA 11+ (e.g. limb injury prevention programs to consider - FIFA 11+ (football/soccer), Footy First (Australian football)).

Materials and Methods

Participants

Page 2, Line 72 – ‘MSC’- please spell out abbreviation

Page 2, Line 73 – should ‘consensus’ be ‘consent’, please consider to be consistent with ethical wording.

Page 2, Line 78 - Please consider tense in expression, recommended should read - Table 1 presents participants characteristics.

Line 83 – Please clarify if the participants performed a run or a sprint? Please ensure this is consistent throughout paper (e.g. Line 26 states run and sprint, is there a difference, this is not clear?)

Line 89-90 - is the following repetitive to the what has already been indicated in this paragraph on Line 83? Suggest the following information on Lines 89-90 could be deleted.

Figure 1 -suggest improvement in made in visual display of flow chart. Also consider if images of exercises/tests be provided, even if this is in an additional supplement. This will assist readers know what exercise players performed.

Results

Table and Figures look useful and results are generally described well and in a succinct way. Please consider the following for improvement.

Line 123 – is the first sentence required here, please consider deleting?

Discussion

 Some important points discussed and linkages to literature made throughout discussion. Limitations appear relevant and noted.

Line 144 – 146 – please consider rewording the expression of this sentence for clarity. Please also refer to other research evidence as suggested previous FIFA 11+ etc. Also consider if this study is related to ‘effectiveness’ or is it ‘efficacy’?

Line 145 – please remove full stop between ‘routines’ and ‘in young…’

Line 155-157 – please improve expression of the following sentence -  The compromised dorsiflexion is strictly related to enhancing of sports injury, noteworthy the alteration of ankle dorsiflexion was demonstrated following ankle damage

Line 159 – please improve expression of the following sentence -  In particular, weight-bearing 158 dorsiflexion is pivotal in a physical activity besides in sports

Line 180-182 - please improve expression of the following sentence - In a recent study, Palazzo and colleagues [56] showed a less decrease of postural control in fatigue condition in the group that used textured insoles inside 181 the athletic shoes, compared with the group that used standard athletic shoes.

Limitations

Line 180-182 - please improve expression of the following sentence - Second, the small number of basketball players participate in this study, further studies should have a larger sample size to confirm our findings.

Please consider suggesting that future studies should also explore that the adoption and maintenance of such exercises in warm up need to be considered in effectiveness trials, including behavioural theory and player and coach perceptions of exercise training (suggest reference - McGlashan and Finch (2010). The extent to which social science theories and models are used in sport injury prevention research, Sports Medicine, 40(10), 841-858.)

Brief conclusion provided, a good finish to the paper. Suggest the implications of the research could be expanded in 1-2 lines.

References

Please check reference style is consistent in reference list.

Author Response

Dear Editor & Reviewers,

First of all, thank you for your comments and suggestions that allowed us to greatly improve the quality of the manuscript. We received by email Reviewer comment 1 and 2 as below. We agree with all your comment, and we corrected point by point the manuscript accordingly. Your comment are in bold text and our responses in plain italics. The modifications highlighted in the original manuscript .Thank you in advance

Reviewer comments 2  : 

 Title

It is recommended that the title of the paper be reworded to be more succinct.

We change the title to

Effectiveness of warm-up routine on the ankle injuries prevention in young female basketball players: a randomized controlled trial.

Abstract

Page 1, Line 19-20 – suggest reword sentence for clarity.

Background/objective: Ankle joint is the most common site of injury in overall athletes including basketball. Thus, continuous examination of effective warm-up (WU) routines in 20 basketball players is a necessity. WU is a period of preparatory exercise to improve training 21 performance and reduce sports injuries.

We change it to

Ankle joint is the most common site of injury for basketball athletes. An effective warm-up (WU) is a period of preparatory exercise to improve training performance and reduce sports injuries. Continuous examination of effective WU routines in basketball players is a necessity.

Page 1, Line 23 – please ensure ROM is spelled out in full with abbreviation for 1st time use.

range of motion (ROM)

Page 1, Line 26 – please reword for clarity “All participants performed 7-min of run and sprint”. Is it All participants performed a 7-min run or sprint? Or is it one or the other? Also please check your reference to 7 mins and then 8 mins on the abstract in correct and also consistent with the methodology section.

All participants performed 7-min of run.

Page 1, Line 29 - Remove the word ‘used’ at the end of the sentence (WU routines were conducted 3 times per week for 10 weeks used.) , or was there something missing at the end of the sentence? Please ensure this reads clearly.

WU routines were conducted 3 times per week for 10 weeks.

Page 1, Line 36 - Conclusions – please add wording ‘are’ to ensure sentence is clear. For example,  Further studies are strongly needed to verify the current findings.

 Further studies are strongly needed to verify our findings.

Introduction

Page 2, Line 41 – suggest add the word ‘for’ to improve expression ‘Ankle sprains account ‘for’ 76.7% of injuries, followed by ankle fractures 16.3%.

Ankle sprains account for 76.7% of injuries

Page 2, Line 47-49 – suggest expression and tense could be improved to enhance clarity and flow of information. Consider the flow in information in introducing the warm up as prevention intervention. Is this sentence placed better in the 2nd paragraph?

The sentence moved to 2nd paragraph

Suggest could say something like the following at the end of the 1st paragraph to improve the significance of the research and highlight why it is important, ‘It is therefore important that ankle injuries are reduced to ensure basketball players can continue to maintain their sport participation and performance’.

Modified within the text

Suggest in the 1st paragraph could also consider the importance of female injury statistics and that they generally have a higher risk of lower limb injuries than males in basketball. Ensure you highlight why did you focus on female athletes?

Basketball has a higher rate of ankle injuries especially between the female athletes [1]

Page 2, Line 57 – change ‘facts’ to ‘fact’.

Changed 

Page 2, Line 59 – change ‘increase’ to ‘increases’

Changed

Page 2, Line 63-65 – please improve expression of the following sentence to ensure clarity - ‘Despite some studies reported an increase in explosive performance was reported after general [27] or specific [28,29] or combined WU [13], others have not reported a significant effect [27,30,31,32].

 Despite some studies reported an increase in explosive performance after general [27], specific [28,29] and combined WU [13], others have not reported a significant effect [27,30,31,32].

Page 2, Lines 65-67 – please consider if this sentence is entirely true and modify accordingly (including references), many lower limb injury prevention programs have been developed worldwide which include ankle joint mobility and balance exercises in other sports and some applied to basketball such as the FIFA 11+ (e.g. limb injury prevention programs to consider - FIFA 11+ (football/soccer), Footy First (Australian football)).

Not considered

Materials and Methods

Participants

Page 2, Line 72 – ‘MSC’- please spell out abbreviation

 Is true name not abbreviation

Page 2, Line 73 – should ‘consensus’ be ‘consent’, please consider to be consistent with ethical wording.

 Consent

Page 2, Line 78 - Please consider tense in expression, recommended should read - Table 1 presents participants characteristics.

 Table 1 presents participants characteristics.

Line 83 – Please clarify if the participants performed a run or a sprint? Please ensure this is consistent throughout paper (e.g. Line 26 states run and sprint, is there a difference, this is not clear?)

 performing sprint with different gaits (7min)

Line 89-90 - is the following repetitive to the what has already been indicated in this paragraph on Line 83? Suggest the following information on Lines 89-90 could be deleted.

Deleted  

Figure 1 -suggest improvement in made in visual display of flow chart. Also consider if images of exercises/tests be provided, even if this is in an additional supplement. This will assist readers know what exercise players performed.

Modified and upload in separated file

Results

Table and Figures look useful and results are generally described well and in a succinct way. Please consider the following for improvement.

Line 123 – is the first sentence required here, please consider deleting?

Deleted

Discussion

Some important points discussed and linkages to literature made throughout discussion. Limitations appear relevant and noted.

Line 144 – 146 – please consider rewording the expression of this sentence for clarity. Please also refer to other research evidence as suggested previous FIFA 11+ etc. Also consider if this study is related to ‘effectiveness’ or is it ‘efficacy’?

To our knowledge, this is the first study evaluated the efficacy of ankle mobility and balance exercises in the warm-up routine on ankle injury prevention in basketball female players.

Line 145 – please remove full stop between ‘routines’ and ‘in young…’

 Modified

Line 155-157 – please improve expression of the following sentence -  The compromised dorsiflexion is strictly related to enhancing of sports injury, noteworthy the alteration of ankle dorsiflexion was demonstrated following ankle damage

 The compromised dorsiflexion is strictly related to reducing of sports injury, noteworthy the alteration of ankle dorsiflexion was demonstrated following ankle damage [38,48].

Line 159 – please improve expression of the following sentence -  In particular, weight-bearing 158 dorsiflexion is pivotal in a physical activity besides in sports

 In particular, dorsiflexion weight-bearing activities is pivotal in a physical activity besides in sports [50].

Line 180-182 - please improve expression of the following sentence - In a recent study, Palazzo and colleagues [56] showed a less decrease of postural control in fatigue condition in the group that used textured insoles inside 181 the athletic shoes, compared with the group that used standard athletic shoes.

In a recent study, Palazzo and colleagues [56] showed a less decrease of postural control in the group that used textured insoles inside the athletic shoes, compared with the group that used standard athletic shoes.

Limitations

Line 180-182 - please improve expression of the following sentence - Second, the small number of basketball players participate in this study, further studies should have a larger sample size to confirm our findings.

Second, the sample size was small, further studies with large sample size are strongly needed to confirm our findings.

Please consider suggesting that future studies should also explore that the adoption and maintenance of such exercises in warm up need to be considered in effectiveness trials, including behavioural theory and player and coach perceptions of exercise training (suggest reference - McGlashan and Finch (2010). The extent to which social science theories and models are used in sport injury prevention research, Sports Medicine, 40(10), 841-858.)

Modified within the text

Brief conclusion provided, a good finish to the paper. Suggest the implications of the research could be expanded in 1-2 lines.

References

Please check reference style is consistent in reference list.

Checked

Round 2

Reviewer 1 Report

Methods. The sample size calculation is missing.

Author Response

Dear Editor & Reviewers,

First of all, thank you for your comments and suggestions that allowed us to greatly improve the quality of the manuscript. We received by email Reviewer comment 1 as below. We agree with all your comment, and we corrected point by point the manuscript accordingly. Your comment are in bold text and our responses in plain italics. The modifications highlighted in the original manuscript. Thank you in advance

Reviewer comments 1  : 

 Methods. The sample size calculation is missing.

The Participants who met the inclusion criteria were 28 participants. They were randomly assigned to either a combined warm-up experimental group (CWU) or general warm-up control group (GWU) using a computer-generated randomization list. In order to prevent any unexpected logistic problems related to the young age, three players previously assigned to a control group was add to the experimental group changing the sample size of both groups (CMU = 17 players, GWU = 11 players).